# Cryo-EM structures of lipidic fibrils of amyloid-β (1-40)

Benedikt Frieg [1,7], Mookyoung Han [2,7], Karin Giller[2], Christian Dienemann [3], Dietmar Riedel[4], Stefan Becker [2] ✉, Loren B. Andreas [2] ✉, Christian Griesinger[2,5] ✉ & Gunnar F. Schröder [1,6] ✉

Alzheimer's disease (AD) is a progressive and incurable neurodegenerative disease characterized by the extracellular deposition of amyloid plaques. Investigation into the composition of these plaques revealed a high amount of amyloid-β (Aβ) fibrils and a high concentration of lipids, suggesting that fibril-lipid interactions may also be relevant for the pathogenesis of AD. Therefore, we grew Aβ40 fibrils in the presence of lipid vesicles and determined their structure by cryo-electron microscopy (cryo-EM) to high resolution. The fold of the major polymorph is similar to the structure of brain-seeded fibrils reported previously. The majority of the lipids are bound to the fibrils, as we show by cryo-EM and NMR spectroscopy. This apparent lipid extraction from vesicles observed here in vitro provides structural insights into potentially disease-relevant fibril-lipid interactions.

Alzheimer's disease (AD) is a progressive and unremitting neurodegenerative disorder characterized by the irreversible loss of memory and cognitive functions (for recent reviews, see refs. [1–4]). The major pathological hallmark of AD is the accumulation and deposition of amyloid-beta (Aβ) peptides (amyloid plaques) outside neurons, followed by the formation of intra-neuronal tangles of the protein Tau, the loss of synaptic function, neuronal death, and irreversible damage of the brain tissue. According to the "amyloid cascade hypothesis"[5–7], AD is related to the production–elimination-imbalance of Aβ peptides, which then initiates the pathology. However, the pathogenesis of AD is still not entirely understood.

Aβ peptides are the product of β- and γ-secretase-mediated cleavage of the amyloid precursor protein[8]. The most abundant variants of Aβ comprise 27–43 amino acids in length. Under pathophysiological conditions, Aβ40 and Aβ42 aggregate into oligomers or arrange into symmetric, periodic fibrils, which have been acknowledged as characteristic features of AD[5,9–11]. Indeed, both Aβ40 and

Aβ42 fibrils were successfully isolated and visualized from AD-diagnosed brain tissue[12,13].

Aβ fibrillization is modulated by multiple factors, including, among a wide range of other factors, Aβ mutations[14], metal ions[15], and lipids[16,17]. The role of lipids in the pathogenesis of AD gained significant attention in recent years[17,18]. For example, studies on the lipid-mediated aggregation of Aβ suggest that negatively charged phospholipids promote Aβ fibrillization, while neutral lipids have little or no effect[19–21]. Additionally, lipid membranes were shown to promote Aβ fibrillization[22,23] and aggregation on the biological membrane, followed by membrane integrity and permeability changes, which are suggested as potential consequences of Aβ-mediated neurotoxicity[19,24–26].

Interestingly, investigations on the composition of Aβ plaques revealed that fibrillar Aβ peptides and lipids colocalize in vivo[27–32], substantiating the role of fibril–lipid interactions in the pathogenesis of AD. However, despite evidence for a significant role of lipids in the Aβ-mediated pathogenesis of AD, very few insights into specific

[1]Institute of Biological Information Processing (IBI-7: Structural Biochemistry) and JuStruct: Jülich Center for Structural Biology, Forschungszentrum Jülich, Jülich, Germany. [2]Department of NMR-Based Structural Biology, Max Planck Institute for Multidisciplinary Sciences, Göttingen, Germany. [3]Department of Molecular Biology, Max Planck Institute for Multidisciplinary Sciences, Göttingen, Germany. [4]Laboratory of Electron Microscopy, Max-Planck-Institute for Multidisciplinary Sciences, Göttingen, Germany. [5]Cluster of Excellence "Multiscale Bioimaging: From Molecular Machines to Networks of Excitable Cells" (MBExC), University of Göttingen, Göttingen, Germany. [6]Physics Department, Heinrich Heine University Düsseldorf, Düsseldorf, Germany. [7]These authors contributed equally: Benedikt Frieg, Mookyoung Han. ✉e-mail: sabe@mpinat.mpg.de; land@mpinat.mpg.de; cigr@mpinat.mpg.de; gu.schroeder@fz-juelich.de

interactions of lipids with Aβ fibrils have been obtained to date. Here we present the cryo-electron microscopy (cryo-EM) structures of six lipidic Aβ40 fibrils that assemble from three filament folds, providing structural insights into fibril–lipid interactions. These stable lipid interactions reveal structural details of lipid-mediated fibrillization relevant to the hypothesis that lipid extraction and disruption of biological membranes play an important role in AD pathology.

## Results and discussion

Human wild-type Aβ40 was recombinantly expressed in *E. coli* and purified to homogeneity. We investigated Aβ40 fibrils formed from this monomeric Aβ40 in the presence of negatively charged liposomes composed of 1,2-dimyristoyl-sn-glycero-3-phosphoglycerol (DMPG) by magic-angle sample spinning NMR spectroscopy and cryo-EM. Initial characterization by negative stain EM revealed fibrils longer than 1 μm, most of them being in contact with spherical or incomplete liposomes attached to the fibril features (Fig. 1a, b). The 3D structure of these lipidic fibrils was solved by cryo-EM, revealing 3 filament folds and 6 distinct fibril polymorphs each of which contains densities assigned to lipids in micelle-like structures. Remarkably, the cross-section of the cryo-EM maps also reveals additional ring- and rod-shaped densities at the fibril surface (e.g., Fig. 2a, b), suggesting that lipid acyl chains are resolved in the cryo-EM maps.

### Fibrillization in the presence of lipids results in a lipidic fibril with a fold similar to brain-seeded fibrils

The L1 fibril, which is most populated in the cryo-EM dataset (Supplementary Fig. 1), is composed of two intertwined protofilaments of the same fold, related by an approximate pseudo 2₁ screw symmetry (Fig. 2a, b; Supplementary Tables 1 and 2). L1 is also dominant in the NMR spectrum (vide infra). Following NMR assignments for 37 residues (Fig. 1c, e; Supplementary Fig. 2; Supplementary Table 3), NOE data (Fig. 3a, b) revealed contacts between the phospholipid acyl chains and residues D23 to G33, the hydrophobic residues L34 and M35 as well as part of the N-terminal β-strand (K16-V18), identifying the additional density in the cryo-EM map of L1 as lipids.

The high-resolution cryo-EM map allows us to accurately model residues D1-V40 (Supplementary Fig. 4a). The L1 fibril is almost identical to previously described Aβ fibrils derived from the brain tissue of an AD patient by seeded fibril growth[33] (Fig. 2c, d). A comparative analysis of $C_\alpha$ and NH chemical shifts showed that 9 of the 21 residues in brain-seeded fibrils had congruent chemical shifts in lipidic fibrils (V18-F20, E22, V24, I32, G33, G37, and K28'). The main difference between the fibrils is that in the brain-seeded fibrils, a proteinaceous density is found approximately where lipid abuts the L1 fibril. This proteinaceous density was proposed to be composed of Aβ40 beta hairpins based on REDOR data and mass-per-length. Furthermore, we observed additional interactions with the acyl chain (Q15-F19) (Fig. 3b) and the head group (D1-H13) (Fig. 2b) of liposomes in the L1 fibril. This analysis indicates that the variations in chemical shifts arise from brain-seeded fibrils interacting with peptides, whereas in our case, fibrils interact with lipids. Furthermore, in contrast to the structure presented here, the N-terminus of the seeded fibrils was not entirely resolved, suggesting that the lipids may be relevant for partial fibril stabilization.

The ring-shaped densities in the cryo-EM cross-sections (Fig. 2a, b) were identified as the lipidic head groups, in line with the findings for lipidic α-synuclein fibrils[34]. The adjacent rod-shaped densities show therefore the lipidic acyl chains[34]. Indeed, hydrophobic surface patches (Supplementary Fig. 5a) are formed by the L1 structure that faces the hydrophobic acyl chains of the lipid molecules. Residues expected to interact favorably with the lipid head groups (Y10, E22, D23, and K28) flank the hydrophobic patches, resulting in the micelle-like arrangement of lipids on the fibril surface, as seen previously for α-synuclein[33] and Aβ oligomers[35]. The cryo-EM map also reveals the periodic arrangement of multiple layers of rod-shaped densities along the helical axis (Supplementary Fig. 6). In contrast, the previous brain-derived structure determined from seeded fibrils revealed, rather than lipids, an additional β-strand-shaped proteinaceous density bound to the hydrophobic patch centered on M35 (Fig. 2c). Lipids were not present in the seeding environment.

The following evidence suggests that L1 also dominates the NMR spectra. First, secondary chemical shifts analyzed by Talos N predict the secondary structure of Aβ40 fibrils to consist of two β-strands ranging from residues H13 to D23 and I31 to V36 that are connected by a loop region. Second, contact between V24 and G33/L34 observed in DARR spectra is consistent with the inter-strand contacts seen in the cryo-EM structure of L1 fibrils (Supplementary Fig. 3). Third, contacts are observed from V24-L34, V24-G33, and S26-D23 that are only compatible with the L1 structure. In the case of the L2 and L3 fibril structures, these inter-residue distances exceed 20 Å (V24-G33, L34) and 8 Å (S26-D23). In the L1 fibril, each layer comprises two molecular entities, leading to a twofold increase in solid-state NMR signal intensity. Conversely, L2 and L3 structures consist of only one molecule per layer. As a result, based on the cryo-EM, L1 fibrils can be expected to account for about 78% of the total signal, while other polymorphs contribute the remaining 22% (15% L2 + L3, 5% L2/L3, 2% L2/L2 + L3/L3).

Based on these signal distribution estimates within the sample and signal-to-noise of about 21 for peaks in the assignment spectra, we conclude that only the L1 polymorph was detectable by NMR. It should be noted that the population estimate from the cryo-EM dataset may not be accurate for several reasons: (1) different polymorphs could react differently to blotting before plunge freezing, (2) automated particle picking might be more effective for the denser L1 fibril, and (3) during image classification only well-defined classes that could be unambiguously identified as one of the polymorphs were selected for further processing.

Consistent with the predominance of one fibril (L1), only one signal set was evident for most residues except for D23 and K28, for which two sets of resonances were assigned. The peak doubling in D23 and K28 indicates structural heterogeneity in the loop within the fibril structure (Fig. 1e; Supplementary Fig. 2). Indeed, according to the molecular model (PDB-ID: 6W0O) previously reported, the salt bridge between these two residues appears to be populated at ~50%, with the remaining 50% being devoid of this salt bridge[33]. This secondary structure, as well as the peak doubling of D23 and K28, match the previously reported brain-derived Aβ40. During molecular dynamics simulations of the L1 fibril, we observed that the hydrogen bond between D23 and K28 breaks but also reforms, with the hydrogen bond being present in 52 ± 1% of all conformations (Supplementary Fig. 7), in agreement with the peak doubling in the NMR experiments.

### Further lipidic structures of Aβ40

L2 and L3 fibrils each consist of a single protofilament and reveal two alternative lipid-induced protofilament folds (Fig. 4; Supplementary Fig. 4b, c). While all 40 amino acids were successfully modeled into the cryo-EM map of the L3 fibril, the N-terminal residues of the L2 fibril are not well resolved, suggesting a higher degree of flexibility in this region. The cross-sections again reveal lipid densities at the fibril surfaces in the neighborhood of hydrophobic residues (Supplementary Fig. 5b, c).

The L2-L3, L2-L2, and L3-L3 fibrils have two intertwined protofilaments (Fig. 5; Supplementary Fig. 5d–f). Interestingly, neither the protofilaments in the L2-L2 nor the L3-L3 fibril form any direct contact. Although both protofilaments in the L2-L3 fibril form some electrostatic interactions, it remains questionable whether these interactions alone would suffice to stabilize the quaternary arrangement. However, for these three fibrils, the cross-sections show lipid densities that mediate interactions between the protofilaments, which are likely essential for stabilizing them.

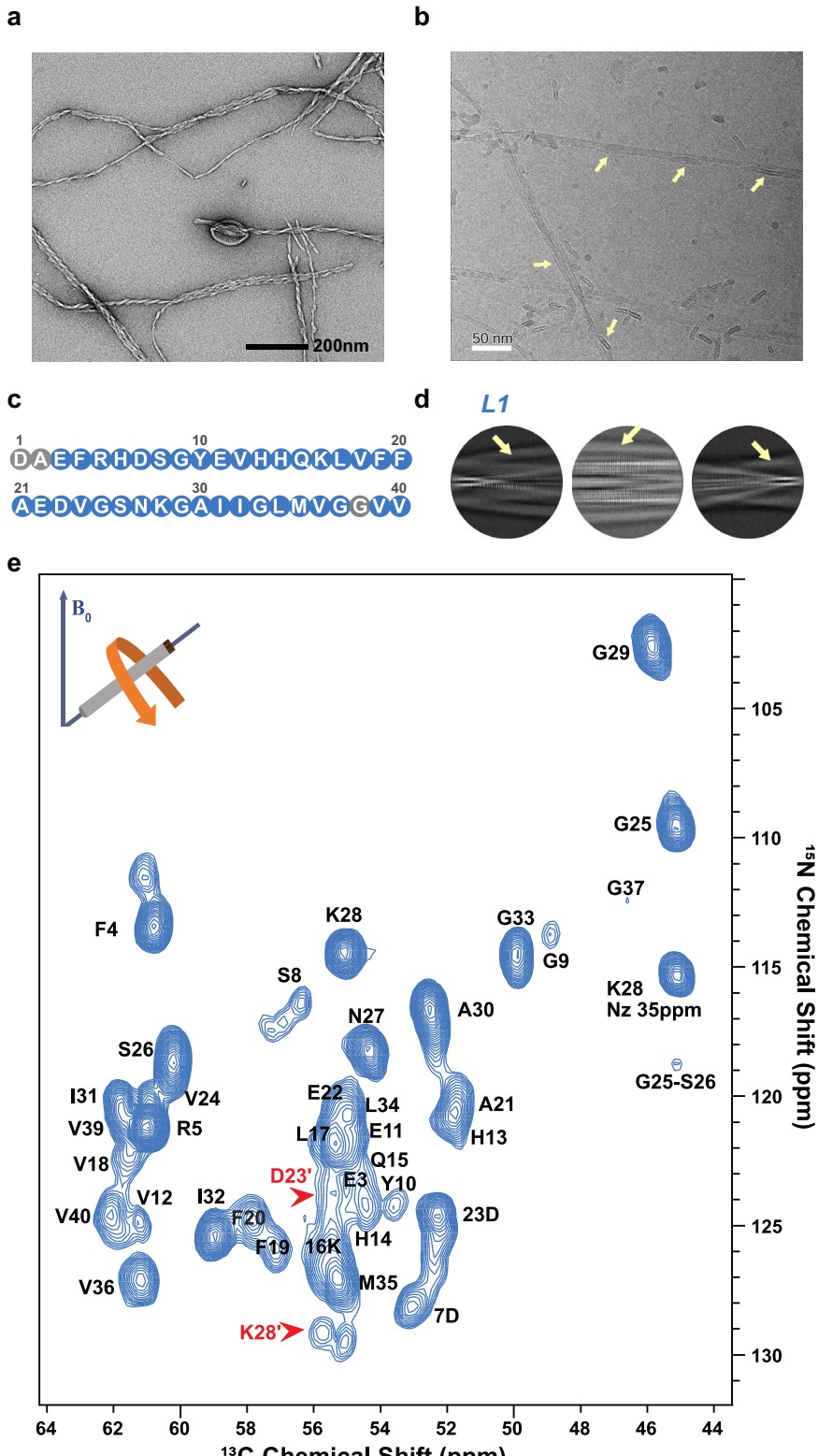

**Fig. 1 | Aβ40 fibrils in the presence of phospholipids. a** A negative stain transmission electron microscopy (TEM) micrograph (a representative from a total of 33). **b** A 20 Å low-pass filtered cryo-electron microscopy (cryo-EM) micrograph of lipidic Aβ40 fibrils (representative from a total of 14,417). The yellow arrows indicate the fibril-bound incomplete liposomes. **c** The sequence of Aβ40. Residues assigned by NMR are colored blue. **d** Examples of 2D class averages for L1. The yellow arrows indicate the fibril-bound layers of lipids (incomplete liposomes), lacking the characteristic amyloid cross-β pattern, while the cross-β pattern is evident for the Aβ40 fibrils. **e** 3D(H)CANH spectrum of lipidic Aβ40 fibrils. Red arrows depict the two populations of K28 and D23.

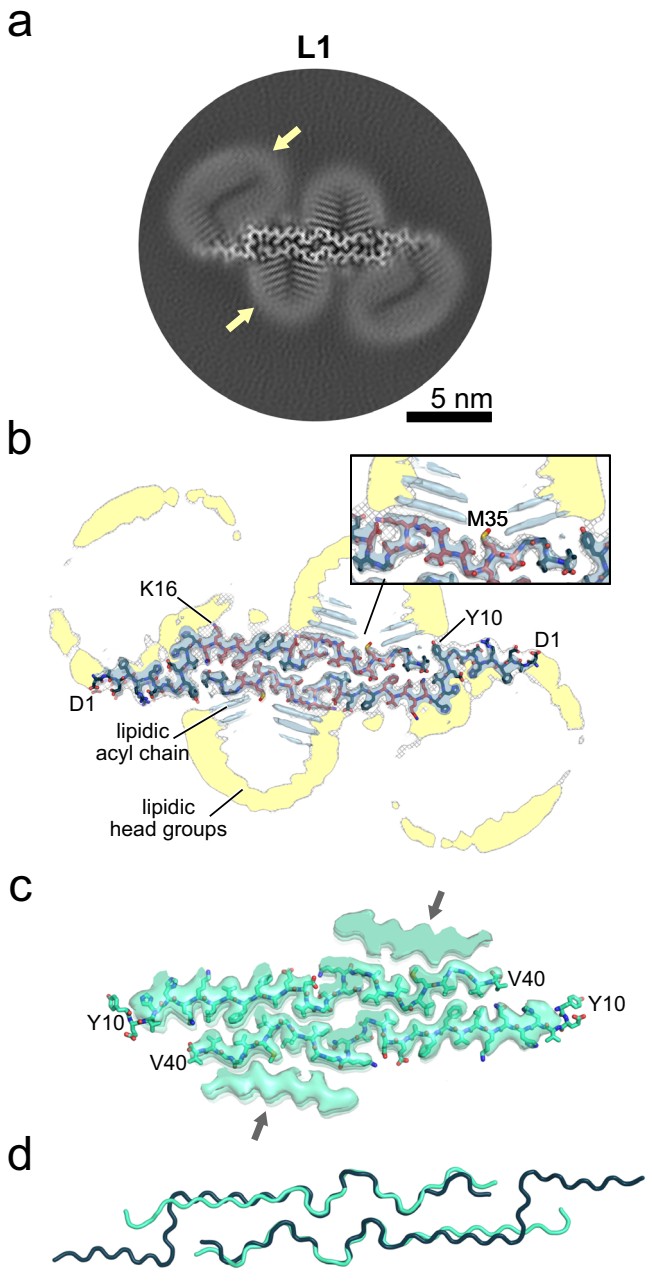

**Fig. 2 | Cryo-EM structures of the lipidic L1 Aβ40 fibril. a** A cross-section of the cryo-EM map, with lipid densities indicated by a yellow arrow. **b** An overlay of a sharpened high-resolution map and an unsharpened, 4.5 Å low-pass-filtered density shown in gray. The atomic model is shown as sticks. Amino acids that interact with lipids in NMR experiments are colored red (Fig. 3). The close-up view shows areas with pronounced lipid densities. Ring-shaped densities in the cryo-EM cross-sections show the lipid head groups, while rod-shaped densities show the lipid acyl chains. Supplementary Fig. 8a shows the sharpened high-resolution maps at different isosurface levels. **c** The sharpened high-resolution map (EMDB 21501) with the docked atomic model (PDB 6W0O) of an Aβ40 fibril derived from the brain tissue of an AD patient and amplified by seeded fibril growth[37]. The arrows indicate proteinaceous densities of unknown sequence[37]. **d** Superposition of the lipidic L1 Aβ40 fibril onto the amplified Aβ40 fibril[37] from panel (**c**).

## Lipidic Aβ fibrils reveal similarities to non-lipidic Aβ fibrils

Although the lipids likely influence the fibril fold, distinct inter- and intra-subunit interactions may also lead to several similarities and dissimilarities between the different lipidic Aβ folds. Indeed, the three lipidic Aβ folds are identical in the central β-strand formed by residues Q15 to A21, with only minor differences in the sidechain rotamers, but they differ in the conformation of the N- and C-termini (Fig. 6a). In particular, the C-terminus adopts a curled, shell-like conformation in both L2 and L3 fibrils, stabilized by interactions between D23 and K28, as well as by hydrophobic intra-subunit interactions (Supplementary Fig. 5b, c). In contrast, the C-terminus in the L1 fibril adopts an extended conformation in which K28 interacts with E22 and D23. Furthermore, interactions between the protofilaments in L1 may also prevent the arrangement towards the curled conformation.

In all lipidic Aβ40 fibrils, interactions between H6, E11, and H13 stabilize the L-shaped N-terminus (Fig. 6a), as seen previously[36]. However, intra-protofilament interactions between Y10 and Q15 in the L3 fibril result in a compact conformation with two consecutive 90° bends of the protein backbone. The absence of this interaction results in a more extended conformation in the L2 fibril. The L1 fibril also reveals a compact N-terminus due to inter-protofilament interactions mediated by V40.

The β-strand formed by $_{15}$QKLVFF$_{20}$ is a common feature in most Aβ fibrils (Fig. 6a–c). This short β-strand seems to be conserved in Aβ40 and Aβ42 and is found in vitro as well as ex vivo[12,13,36,37]. Previous studies identified the hydrophobic regions $_{17}$LVFFA$_{21}$ and $_{30}$AIIGLM$_{35}$ as well as $_{41}$IA$_{42}$ for Aβ42 as most important for aggregation and neurotoxicity[34,38]. In particular, interactions between F19 and L34 are suggested to be essential for the cellular toxicity of Aβ40[39]. Furthermore, such interactions are present in the lipidic L1 fibril and in both ex vivo Aβ42 fibrils[12], supporting the disease relevance of the lipidic fibril L1.

## Lipid-mediated Aβ fibrillization leads to lipid vesicle disruption and remodeling

In the present study, we utilized a mildly acidic pH of 6.5 to facilitate partial protonation of the three histidine residues in the monomeric, non-fibrillar Aβ40 peptide. This facilitates enhanced electrostatic interactions between the Aβ40 monomer and negatively charged head groups of liposomes and promotes Aβ40 aggregation. For the study, DMPG was selected as the lipid due to its transition temperature ($T_m$) of 24 °C, approximating the physiological conditions experienced by neuronal cell membranes at our designated incubation temperature of 37 °C.

For the aggregation studies, liposomes were prepared using a sonication method. Dynamic light scattering (DLS) measurements confirmed that the average diameter of these liposomes exceeded 80 nm. After aggregation, we observed the presence of spherical-shaped liposomes on the fibrils, each with an approximate diameter of 100 nm (Fig. 1a), as evidenced by negative-staining electron microscopy (EM) data. It is noteworthy that during the aggregation process, no sonication was applied; thus, any observed lipid assembly with a diameter less than 80 nm can be attributed to their interaction with aggregating Aβ40.

In our density map analysis, we observed lipid micelles on the Aβ40 fibril molecule with a diameter of approximately 10 nm, smaller or similar to an individual Aβ40 fibril molecule (dimensions L1: 13.4 nm, L2: 5.1 nm, L3: 6.6 nm). In the case of L1, extra density is located between Lys28 and Val40, extending over 3.3 nm, corresponding to the combined length of two acyl chains in a DMPG lipid molecule (Fig. 2a, b). This observation is corroborated by the NOE data (Fig. 3b). Altogether, we propose that during Aβ40 aggregation, lipids interact with Aβ40 and form such lipid micelles on the surface of the fibrils.

It is conceivable that the process observed here in vitro happens in a similar way also in vivo. This would lead to destabilization, reduction of structural integrity and permeability of cell membranes[24,40], which eventually leads to cell death[25,41]. Indeed, this

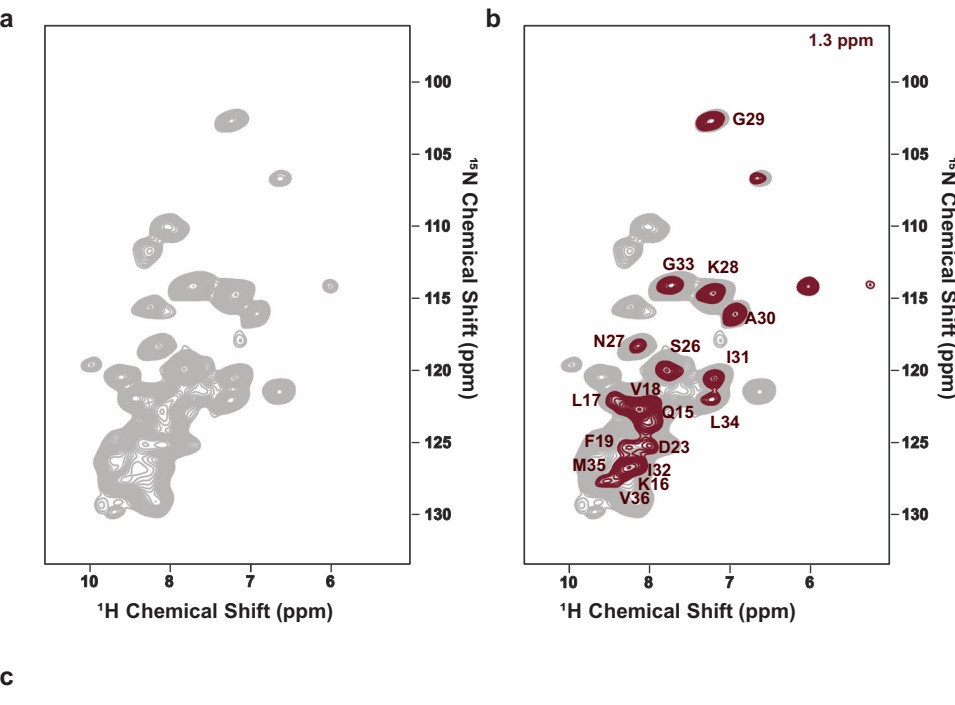

**Fig. 3 | Backbone and side-chain spectrum of Aβ40 fibrils in the presence of phospholipids. a** 2D (H)NH spectrum of the $^2H^{15}N^{13}C$ lipidic Aβ40 fibril (gray). **b** the plane of a 3D H(H)NH spectrum in 50 ms NOE mixing corresponding to lipid-CH$_2$ close contact with the fibril (1.3 ppm, red). **c** Sequence and secondary structure of human Aβ40. Red-colored residues bind to the lipid acyl chain in NMR experiments.

process was already discussed in the context of neurotoxicity[24,40,42,43] and also for other peptides[44–46].

In conclusion, we determined 3D structures of Aβ40 fibrils bound to lipid molecules. The complex structures provide a molecular-level structural understanding of how Aβ aggregation can lead to lipid extraction from vesicles, which is considered to play a critical role in current pathogenesis models of AD.

## Methods
### Protein expression and purification
Uniformly $^{15}N,^{13}C$-labeled recombinant Aβ40 was produced as follows: Aβ40 was expressed as a SUMO fusion protein using a modified pET32a vector (Novagen). $^2H,^{13}C,^{15}N$-labeled Aβ40 was expressed in *E. coli* strain BL21(DE3) adapted to 100% $^2H_2O$ minimal medium supplemented with $^2H_7,^{13}C_6$-β-D-glucose and $^{15}N$-ammonium chloride. Protein from a 1 l expression culture was solubilized in 20 mM Tris pH 7.5, 200 mM NaCl, 10 mM imidazole, complete EDTA free, 0.5 mM PMSF and purified by immobilized metal affinity chromatography using a 5 ml nickel column (Macherey-Nagel). The protein was eluted from the nickel column with a solubilization buffer supplemented with 500 mM imidazole. The fusion protein was dialyzed (Biotech CE tubing, MWCO: 8–10 kDa, Spectra/Por) against Tabacco Etch Virus (TEV) digestion buffer (50 mM Tris pH 7.8, 5 mM imidazole, 0.5 mM EDTA, 0.5 mM PMSF, 1 mM DTT) and digested overnight on ice with TEV protease (3 mg TEV protease/100 mg SUMO-Aβ40 fusion protein). After TEV digestion the protein solution was supplemented with 6 M guanidinium hydrochloride, and the SUMO protein was mostly removed by several passages through two concatenated 5 ml nickel columns. Final purification was performed on a C4 reversed phase Vydac HPLC column. Aβ40 peptide eluted in a linear (0–100%) acetonitrile gradient from this column as a single peak. The purified peptide was lyophilized before use.

### Lipidic Aβ40 fibril preparation
Vesicles were prepared from 1,2-dimyristoyl-sn-glycero-3-phosphoglycerol (DMPG) film prepared by dissolution in chloroform: methanol (ratio 2:1) and subsequent evaporation of the solvent under an $N_2$-stream and lyophilization. The DMPG lipid film was sonicated for 5 min in 10 mM Na-P$_i$ buffer at pH 6.5. Aβ40 was dissolved in 0.1 M NaOH for 30 min at room temperature with 2 mM as the final stock concentration. The Aβ40 stock was diluted in 10 mM Na-P$_i$ at pH 6.5 with the vesicles to a final concentration of 20 μM protein and 600 μM lipid. The sample was incubated under quiescent conditions for 1–2 days at 37 °C until fibrils formed. The aggregation process was monitored by mixing Thioflavin T-containing buffer (10 μM ThT, 10 mM Na-P$_i$ at pH 6.5) and measuring the fluorescence emission intensity at 482 nm in a Varian Cary Eclipse fluorescence spectrometer.

### Solid-state NMR measurements
For the backbone assignment and determination distance between lipid and fibril experiment, we used the same fibril formation protocol on $^2H^{13}C^{15}N$ Aβ40 fibril and the $^{13}C^{15}N$ Aβ40 fibril (2D $^{13}C$–$^{13}C$ DARR, cryo-EM, and (h)NCA). And spectrum similarity was checked by (h)NCA on the $^{13}C^{15}N$ Aβ40 fibril and (h)CANH on the $^2H^{13}C^{15}N$ Aβ40 fibril. The 2D (h)NH, 3D (h)CANH, (h)coCAcoNH, (h)CONH, (h)COcaNH, (h)caCBcaNH, and (h)caCBcacoNH experiments for protein assignments, and the 3D H(h)NH (NOE, 50 ms) were acquired on the $^2H^{13}C^{15}N$ Aβ40 fibril (the 800-MHz Bruker Avance III HD spectrometer at a magnetic field of 18.8 T equipped with a 1.3-mm magic-angle spinning (MAS) HCN probe and MAS at 55 kHz and temperature 235 K)[41].

Chemical shift data for $^{13}CO$, $^{13}C_\alpha$, and $^{13}C_\beta$ obtained from sequence assignment spectra were used in TALOS-N to obtain predictions on secondary structure as well as dihedral backbone angles[47]. All 2D $^{13}C$–$^{13}C$ DARR and (h)NCA were acquired on the 850-MHz Avance III and NEO spectrometer with a 3.2-mm MAS HCN probe at a magnetic

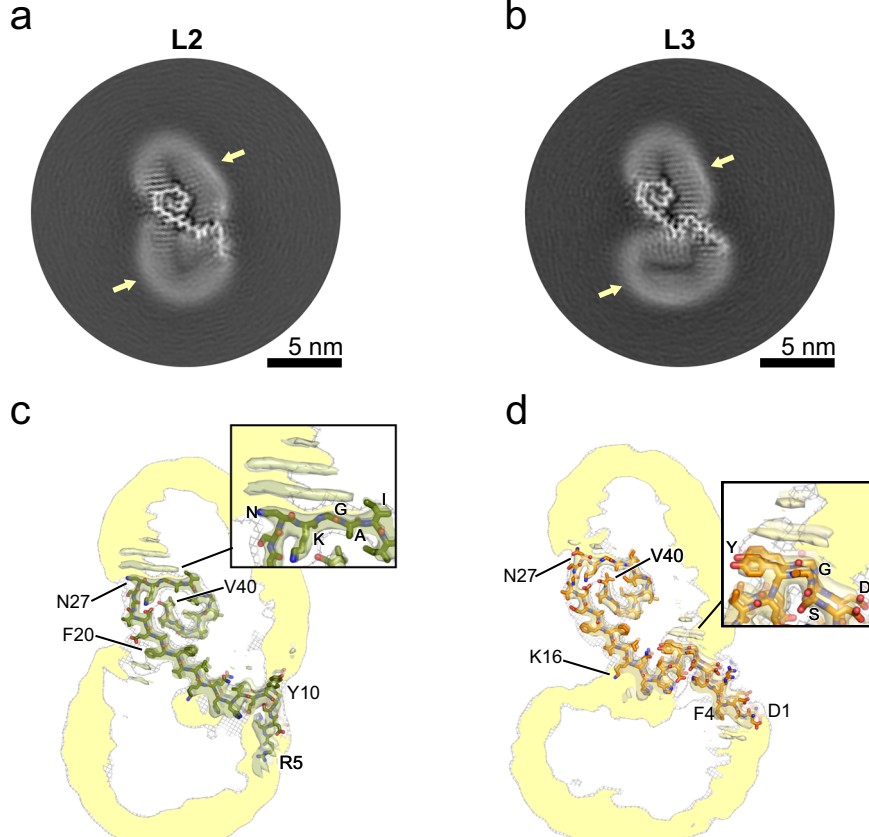

**Fig. 4 | Cryo-EM structures of lipidic L2 and L3 Aβ40 fibrils. a, b** Cross-sections of the cryo-EM maps of the L2 (**a**) and L3 (**b**) Aβ40 fibrils, with lipid densities indicated by a yellow arrow. **c, d** Overlay of a sharpened high-resolution map and an unsharpened, 4.5 Å low-pass filtered density is shown in gray for the L2 (**c**) and L3 (**d**) Aβ40 fibrils. The close-up view shows areas with pronounced lipid densities, with ring-shaped densities in the cryo-EM cross-sections showing the lipid head groups, while the rod-shaped densities show the lipid acyl chains. Supplementary Fig. 8b, c shows the sharpened high-resolution maps at different isosurface levels.

field of 20.0 T and MAS at 17 kHz (265 K). NMR experimental parameters are reported in Supplementary Table 4. NMR chemical shift dates are deposited at the Biological Magnetic Resonance Data Bank with entry BMRB 52006.

### Transmission electron microscopy

Samples were bound to glow-discharged carbon foil-covered 400 mesh copper grids. Samples were stained using NanoVan (Nanoprobes Inc.) and evaluated at room temperature using a Talos L120C (Thermo Fisher Scientific).

### Cryo-EM grid preparation and imaging

For cryo-EM grid preparation, 1.5 µL of fibril solution was applied to freshly glow-discharged R2/1 holey carbon film grids (Quantifoil). After the grids were blotted for 12 s at a blot force of 10, the grids were flash-frozen in liquid ethane using a Mark IV Vitrobot (Thermo Fisher).

Cryo-EM data sets were collected on a Titan Krios transmission electron microscope (Thermo Fisher) operated at 300 keV accelerating voltage and a nominal magnification of ×81,000 using a K3 direct electron detector (Gatan) in non-super-resolution counting mode, corresponding to a calibrated pixel size of 1.05 Å. Data acquisition was done in EFTEM mode using a Quantum LS (Gatan) energy filter set to a slit width of 20 eV. A total of 14,417 movies were collected with SerialEM[48]. Movies were recorded over 40 frames accumulating a total dose of ~40.5 e⁻/Å². The range of defocus values collected spans from −0.7 to −2.0 µm. Collected movies were motion-corrected and dose-weighted on the fly using Warp[49].

### Helical reconstruction of Aβ fibrils

Aβ fibrils were reconstructed using RELION-3.1[50], following the helical reconstruction scheme[44]. Firstly, the estimation of contrast transfer function parameters for each motion-corrected micrograph was performed using CTFFIND4[45]. Next, filament picking was done using crYOLO[46].

For 2D classification, we extracted 3,384,825 particle segments using a box size of 600 pix (1.05 Å/pix) downscaled to 200 pix (3.15 Å/pix) and an inter-box distance of 13 pix. L1 and L2–L3 fibrils were separated at this 2D classification stage, whereas L2 and L3 as well as L2–L2 and L3–L3 were too similar on the 2D level (Supplementary Fig. 1b).

For 3D classification, the classified segments after 2D classification were (re-)extracted using a box size of 250 pix (1.05 Å/pix) and without downscaling. Starting from a featureless cylinder filtered to 60 Å, several rounds of refinements were performed while progressively increasing the reference model's resolution. The helical rise was initially set to 4.75 Å and the twist was estimated from the micrographs. Once the β-strands were separated along the helical axis, we optimized the helical parameters (final parameters are reported in Supplementary Table 1). After multiple rounds of focused 3D classification on the Aβ regions of the lipidic fibrils with 3–5 classes, we successfully separated L2 and L3 as well as L2–L2 and L3–L3, which were then treated individually. For all fibrils, we then performed a 3D classification with only a single class and without focusing, followed by gold-standard 3D auto-refinement. Standard RELION post-processing with a soft-edged solvent mask that includes the central 10% of the box height yielded post-processed maps (B-factors are reported in Supplementary

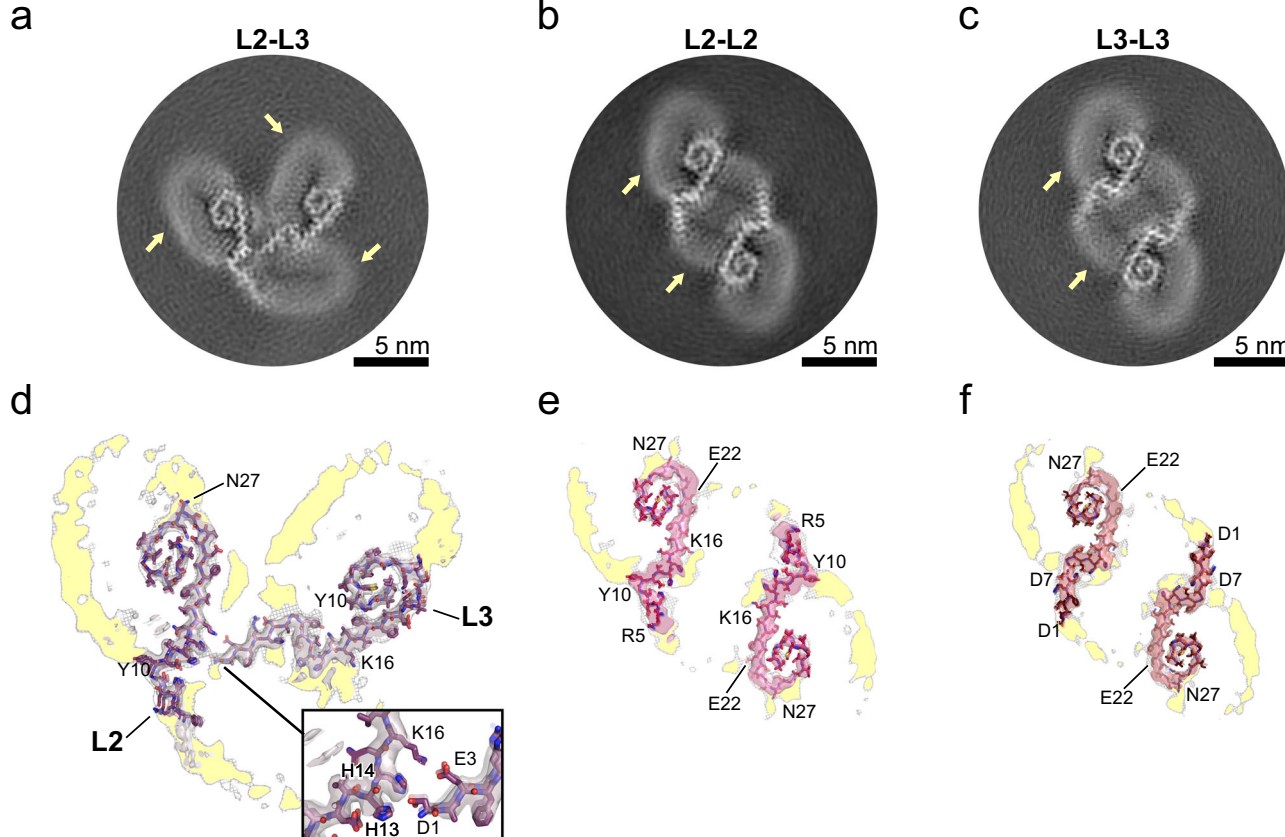

**Fig. 5 | Cryo-EM structures of lipidic L2–L3, L2–L2, and L3–L3 Aβ40 fibrils.**
**a–c** Cross-sections of the cryo-EM maps of the L2–L3 (**a**), L2–L2, (**b**), and L3–L3 (**c**) Aβ40 fibrils, with lipid densities indicated by a yellow arrow. **d–f** Overlays of a sharpened high-resolution map and an unsharpened, 4.5 Å low-pass filtered density are shown in gray for the L2–L3 (**d**), L2–L2, (**e**), and L3–L3 (**f**) Aβ40 fibrils. The close-up view shows areas with pronounced lipid densities, with ring-shaped densities in the cryo-EM cross-sections showing the lipid head groups, while the rod-shaped densities show the lipid acyl chains. Supplementary Fig. 8d–f shows the sharpened high-resolution maps at different isosurface levels.

Table 1). The resolution was estimated from the value of the FSC curve for two independently refined half-maps at 0.143 (Supplementary Fig. 9). Density maps colored according to local resolution are shown in Supplementary Fig. 14. The optimized helical geometry was then applied to the post-processed maps yielding the final maps used for model building. For the L1 fibril, a left-handed twist was visible in the final cryo-EM map (Supplementary Fig. 10), whereas the resolution was not sufficient for the other fibrils, such that also a left-handed twist was assumed.

## Atomic model building and refinement

For the Aβ40 L1 fold, one protein chain was extracted from PDB-ID 6W0O[37] of wild type Aβ40, and the N-terminal region D1-G9 was built de novo in Coot[51]. The atomic model of the Aβ40 L2 and L3 fibrils was built de novo in Coot[51], which later served as starting models for the L2–L3, L2–L2, and L3–L3 fibrils. Subsequent refinement in real space was conducted using PHENIX[52,53] and Coot[51] in an iterative manner. The resulting models were validated with MolProbity[54] and details about the atomic models are described in Supplementary Table 2.

## Molecular dynamics simulations of the L1 Aβ 40 fibril

To investigate the intra-molecular hydrogen bond between D23 and K28, we performed unbiased molecular dynamics (MD) simulations of the L1 Aβ40 fibril. Starting from our cryo-EM structure, we built a model composed of 24 helically arranged peptide chains. Residue protonation states were assigned for a pH of 7 and hydrogen atoms were added to the cryo-EM structure according to the Amberff19SB force field library[55] by using LEaP[56], which is distributed with the Amber

22 suite of programs (comprised of AmberTools22 and Amber22)[57], such that all glutamate and aspartate residues are negatively charged, lysine and arginine positively charged, and tyrosine and histidine neutral. After adding 72 sodium ions to neutralize the system, we placed the fibril-ion complex into a truncated-octahedron solvent box leaving at least 15 Å between any solute atom and the edge of the simulation box. The Amber ff19SB force field[55] was applied to describe the Aβ40 fibril. Ion parameters for sodium ions were taken from ref. 58 and used in with the OPC water model[59].

The exact minimization, thermalization (towards 300 K), and density adaptation (towards 1 g/cm³) protocol are reported in ref. 60, which was applied previously to study amyloid fibrils[61,62] as well as lipidic amyloid fibrils[33]. The conformation after thermalization and density adaptation served as starting points for the subsequent ten NPT production simulations. Therefore, we (re-)started ten independent NPT production simulations at 300 K and 1 bar for 1 μs each, in which new velocities were assigned from Maxwell–Boltzmann distribution during the first step of the NPT production simulation. However, without the final proper arrangement of lipids around the fibrils we observed that the N- and C-termini, as well as the top and bottom layers of the fibril, are highly mobile (Supplementary Figs. 11a, b and 12a). Hence, we weakly (0.1 kcal mol⁻¹ Å⁻²) restrained the $C_\alpha$ atoms to the initial atomic coordinates, as described previously[33], which was enough to preserve the L1 fold also at the termini (Supplementary Figs. 11c and 12b). Important to note, that all non-$C_\alpha$ atoms were allowed to move freely.

During production simulations, Newton's equations of motion were integrated in 4 fs intervals, applying the hydrogen mass

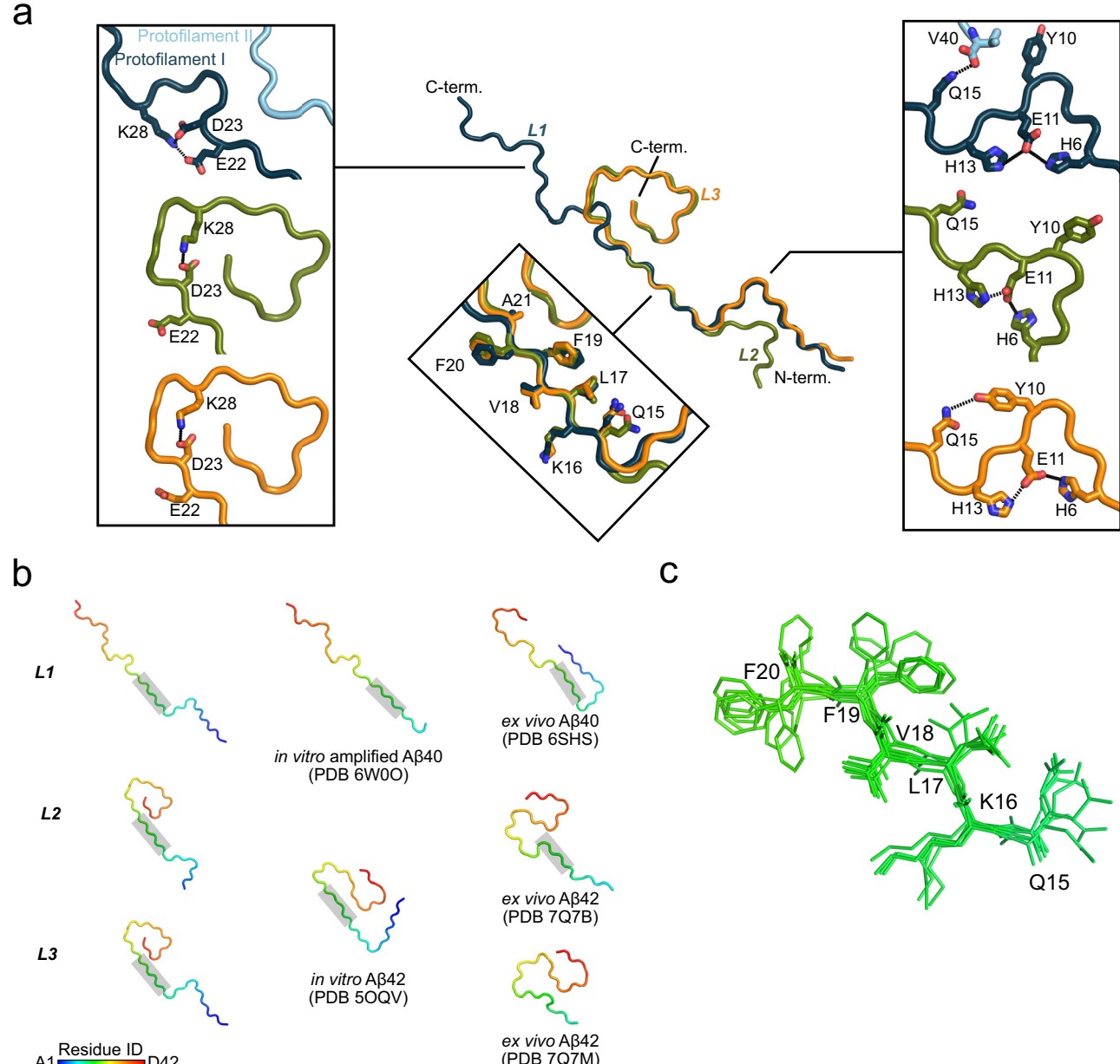

**Fig. 6 | Comparison of the lipidic Aβ40 folds with other known structures.**
**a** Superposition of a single protein chain of L1 (blue: protofilaments I and II in different shades of blue), L2 (green), and L3 (orange) Aβ40. The close-up views show regions of similarities and dissimilarities between the three folds. Interactions between residue side chains (shown as stick-model) are depicted as dashed lines. **b** Backbone traces of the lipidic Aβ40 folds and previously resolved Aβ40[13,37] and Aβ42[12,36] folds, with residues colored according to the rainbow palette in the lower left corner. Residues with a gray background share a high similarity and the superposition is shown in (**c**).

repartitioning approach[63] to all non-water molecules, which were handled by the SHAKE algorithm[64]. Coordinates were stored in a trajectory file every 200 ps. The minimization, thermalization, and density adaptation were performed using the pmemd.MPI[65] module from Amber22[57], while the production simulations were performed with the pmemd.CUDA module[66].

We used cpptraj[67] from Amber22[57] to analyze the trajectories. Hydrogen bond interactions between D23 and K28 were determined using a distance of 3 Å between the donor and acceptor heavy atoms and an angle (donor atom, H, acceptor atom) of 135° as cutoff criteria. Considering only the central 12 Aβ40 chains, we then calculated a fraction for each interaction pair in each MD trajectory when the hydrogen bond is formed. Finally, we calculated the average fraction ± SEM (standard error of the mean) of the D23-K28 hydrogen bond

occurrence over all 10 replica simulations ($n = 120$). We also calculated the autocorrelation function of the hydrogen bond interactions between D23 and K28, for each replica simulation individually (Supplementary Fig. 13a) but also for the cumulative trajectory ($10 \times 1\,\mu s$) (Supplementary Fig. 13b).

For the hierarchical agglomerative clustering of the non-restrained trajectories, the distance between the coordinate frames was calculated via a best-fit coordinate RMSD considering all C atoms. The clustering was finished when the minimum distance $\varepsilon$ between clusters was > 4 Å.

### Reporting summary
Further information on research design is available in the Nature Portfolio Reporting Summary linked to this article.

## Data availability

Cryo-EM maps have been deposited in the Electron Microscopy Data bank (EMDB) under the accession numbers (L1) EMD-17218, (L2) EMD-17223, (L3) EMD-17234, (L2–L3) EMD-17235, (L2–L2) EMD-17238, and (L3–L3) EMD-17239. The corresponding atomic models have been deposited in the Protein Data Bank (PDB) under the accession numbers: 8ovk (L1), 8ovm (L2), 8owd (L3), 8owe (L2–L3), 8owj (L2–L2), and 8owk (L3–L3). NMR chemical shift dates are deposited at the Biological Magnetic Resonance Data Bank with entry BMRB 52006. Source data are provided with this paper.

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

## Acknowledgements

This work was supported by the Max Planck Society (to C.G.) and the Deutsche Forschungsgemeinschaft (DFG, German Research Foundation) under Germany's Excellence Strategy-EXC 2067/1-390729940 (to C.G.) and the Emmy Noether program to LBA (project number: 397022504). B.F. and G.F.S. are grateful for the computational support and infrastructure provided by the "Zentrum für Informations-und Medientechnologie" (ZIM) at the Heinrich Heine University Düsseldorf and the computing time provided by Forschungszentrum Jülich on the supercomputer JURECA-DC at Jülich Supercomputing Centre (JSC).

## Author contributions

C.G., L.B.A., S.B., and G.F.S. designed and supervised the project. B.F., M.H., C.G., and G.F.S. administered the project. K.G. and S.B. performed protein expression and purification. M.H. prepared the fibril samples. M.H. and L.B.A. performed NMR experiments. D.R. optimized the sample quality screening conditions using negative stains. C.D. prepared the cryo-EM grids and collected the cryo-EM images. B.F. processed the cryo-EM images, reconstructed the fibril structures, and built the atomic models. B.F. and M.H. visualized the results. The original draft was written by B.F., M.H., S.B., L.B.A., C.G., and G.F.S. All authors reviewed and edited the manuscript.

## Funding

## Competing interests

B.F. is now an AstraZeneca employee. The other authors declare no competing interests.
