## [Peer review file · Nature Communications]

REVIEWER COMMENTS

Reviewer #1 (Remarks to the Author):

This is an excellent manuscript that reports the results of cryoEM and solid state NMR measurements on 40-residue amyloid-beta (Ab40) fibrils prepared in vitro in the presence of phospholipid vesicles. The authors find a predominant fibril structure (L1) that closely resembles a structure reported previously in studies of brain-derived fibrils by Ghosh et al., but with lipids bound to the fibril surface in place of extra layers of peptide molecules. This result suggests that the L1 structure is relevant to Alzheimer's disease, and that the structure reported by Ghosh et al. may be a consequence of fibril amplification in the absence of lipids. Amazingly, the authors see ordered lipid molecules adjacent to fibril surfaces in their 2D class average images. Additionally, the authors report L2 and L3 structures that are qualitatively different from any amyloid-beta fibril structures that have been described previously, either from solid state NMR or cryoEM studies. Overall, this is an important contribution to our understanding of amyloid-beta fibril formation and of interactions between fibrils and lipids (which may be an important mechanism for neurotoxicity).

The authors should address the following minor points in order to clarify various aspects of the presentation:

1. My copy of the manuscript has two figures that are both labeled Figure 1. The authors should fix this and check that they are referring to the correct figure numbers throughout the text.
2. In the first paragraph of the results section, the authors should state that they are using recombinant Ab40, with the wild-type human sequence and without additional amino acids or tags.
3. How do the solid state NMR spectra of these "lipidic" fibrils compare with solid state NMR spectra of brain-seeded fibrils reported previously by Ghosh et al.? Are the same chemical shifts observed? The spectra of "lipidic" fibrils are apparently better-resolved. Is this because signals from the outer "hairpin" layers in the brain-seeded fibrils of Ghosh et al. are absent from the lipidic fibrils, or because the brain-seeded fibrils were more highly polymorphic?
4. The authors say that the D23-K28 salt bridge "was previously reported...to be populated to around 50%", and they cite the paper by Ghosh et al. I think this is incorrect. The fsREDOR data in the paper by Ghosh et al. seem to indicate nearly full occupancy of the D23-K28 salt bridge.

5. According to Fig. S1, the L1 fibrils represent 63% of the particles. The solid state NMR spectra indicate a higher level of structural homogeneity. Why? Is this because the relative numbers of cryoEM particles are not an accurate representation of the relative quantities (i.e., masses) of different polymorphs? Or is this because the solid state NMR samples were really more homogeneous than the cryoEM samples? Were the same batches of fibril used for both cryoEM and solid state NMR?

6. Another possibility is that the L2 and L3 fibrils are less stable thermodynamically than the L1 fibrils, so that the L2 and L3 fibrils could convert slowly to L1 fibrils during or prior to solid state NMR measurements. Did the authors investigate this possibility? Perhaps they should at least mention that the different polymorphs are likely to have different stabilities. I would guess that the L1 fibrils are more stable than other polymorphs, due to their more extensive intermolecular contacts.

Reviewer #2 (Remarks to the Author):

In this manuscript, the authors report the polymorphic structures of A β 40 fibrils grown with lipid membranes by using cryo-EM. They provide detailed structural information about the fibrils and compare them with previously reported A β 40 fibrils conformations. As lipid molecules are considered critical factors to modulate protein aggregation, this study will provide insights into protein-lipid interactions during amyloid fibril formation, but some issues listed below should be addressed before publication.

1. Cellular membranes are formed by various lipids, but in this study, the authors prepared the vesicles using a saturated and negatively charged DMPG lipid. Why did the authors choose DMPG as a model system to study the fibril-lipid interactions? Are there any specific reasons for this, such as the important physiological roles of DMPG in Alzheimer's disease and Ab aggregation?

2. The authors mentioned that "Initial characterization by negative stain EM revealed fibrils longer than 1 μ m, most of them being in contact with spherical or incomplete fibril-attached features (Fig. 1a, b)". However, it is hard to find what are the incomplete fibril-attached features in the figure, so it would be good to add arrows to indicate these features. And also please add descriptions about yellow arrows in Fig 1b.

3. The samples are prepared by mixing Ab proteins with lipid vesicles, but negative stained EM and raw micrograph images only showed the fibril without lipid vesicles. Did authors observe vesicle-associated fibrils or broken vesicles by fibrils from negative stained EM or cryo-EM images? Since the reconstructed 3D density maps show lipid features, there are probably cryoEM images that show membrane-associated fibrils. If yes, please show the images. I think this will help readers to understand how lipid vesicles interact with fibrils.

4. In this study, cryo-EM report three different folds, and the secondary structure of L1 significantly differs from L2 or L3. For example, the C-terminal region of L1 is different from the same region of L2

and L3 polymorphs. Therefore, readers may expect the heterogeneous ssNMR spectra (multiple peaks) because chemical shifts of NMR spectra are known to be very sensitive to the secondary structure of the protein. However, the peaks in the ssNMR spectra are very homogeneous except for residues D23 and K28. Can the author provide more explanation about this, why signals from L2 and L3 polymorphs are invisible in ssNMR spectra except D23 and K28?

5.L2 fold has a fibril core from residue 5 to residue 40, which means there is a somewhat mobile N-terminal region (residue 1-4). Have the authors tried to measure the dynamic signals by NMR?

6.The figure number should be corrected. There are two Fig. 1 on pages 18 and 19, so all figure numbers are not matched with the main text.

7.Reference #36 page 4 seems wrong. I think the author wanted to refer to #34, not #36. However, the authors are correct and intend to refer to #36, disregard my comment.

8.In conclusion, the authors claim that “ The complex structures provide a molecular-level structural understanding of how A β aggregation can lead to lipid extraction from vesicle”. How do authors conclude that lipids associated with fibrils in density maps are extracted from vesicles? I think sonicated vesicles may be simply attached to the fibrils. Are there any further data to support lipid extraction from vesicles?

Reviewer #3 (Remarks to the Author):

The authors have reported cryo-EM structures of amyloid-beta(1-40) fibrils in lipid vesicles, which might be useful in investigating fibril-lipid interactions present in Alzheimer’s patients. The authors have also used MD simulations to investigate hydrogen bond interactions between D23 and K28. I only have the following questions regarding the MD simulations for the authors:

1. Please state the pH (this is usually set with PROPKA or H++ programs) and the ionic concentration that were used for the MD simulations. I’m a bit concerned that the right pH and ionic concentrations were not set since they were not stated in the methods. Please also state whether a cubic or octahedral box was used for the solvent.
2. Ten simulations with 1 microsecond long each seem to be long enough but it would be better to justify the simulation length as done in the SI of Robustelli et al. JACS 2022 (i.e., find the autocorrelation times).
3. It’s unclear how the conformations were chosen and extracted for Fig S11. Was a clustering method used or were these final snapshots at 1 microsecond for each of the ten simulations? It would be more informative if a clustering analysis was done to show that these certain highly mobile or immobile conformations are the dominant conformations depending on the restraints.

REVIEWER COMMENTS

Reviewer #1 (Remarks to the Author):

This is an excellent manuscript that reports the results of cryoEM and solid state NMR measurements on 40-residue amyloid-beta (Ab40) fibrils prepared in vitro in the presence of phospholipid vesicles. The authors find a predominant fibril structure (L1) that closely resembles a structure reported previously in studies of brain-derived fibrils by Ghosh et al., but with lipids bound to the fibril surface in place of extra layers of peptide molecules. This result suggests that the L1 structure is relevant to Alzheimer's disease, and that the structure reported by Ghosh et al. may be a consequence of fibril amplification in the absence of lipids. Amazingly, the authors see ordered lipid molecules adjacent to fibril surfaces in their 2D class average images. Additionally, the authors report L2 and L3 structures that are qualitatively different from any amyloid-beta fibril structures that have been described previously, either from solid state NMR or cryoEM studies. Overall, this is an important contribution to our understanding of amyloid-beta fibril formation and of interactions between fibrils and lipids (which may be an important mechanism for neurotoxicity).

The authors should address the following minor points to clarify various aspects of the presentation:

1. My copy of the manuscript has two figures that are both labeled Figure 1. The authors should fix this and check that they are referring to the correct figure numbers throughout the text.

Thank you for bringing this to our attention. We regret the oversight and we have corrected the figure numbers.

2. In the first paragraph of the results section, the authors should state that they are using recombinant Ab40, with the wild-type human sequence and without additional amino acids or tags.

We appreciate your suggestion and have added information regarding the wild-type A β 40 sequence in the first paragraph and that it has been recombinantly expressed. In the revised manuscript we added the paragraph to the Results and Discussions on page 4, starting at line 3. In the revised manuscript we now write:

"Human wild-type A β 40 was recombinantly expressed in *E. coli* and purified to homogeneity. We investigated A β 40 fibrils formed from this monomeric A β 40 in the presence of negatively charged liposomes composed of 1,2-dimyristoyl-sn-glycero-3-phosphoglycerol (DMPG) by magic-angle sample spinning NMR spectroscopy and cryo-EM. Initial characterization by negative stain EM revealed fibrils longer than 1 μ m, most of them being in contact with spherical or incomplete liposomes attached to the fibril features (Fig. 1a, b)."

3. How do the solid-state NMR spectra of these "lipidic" fibrils compare with solid state NMR spectra of brain-seeded fibrils reported previously by Ghosh et al.? Are the same chemical shifts observed? The spectra of "lipidic" fibrils are apparently better resolved. Is this because signals from the outer "hairpin" layers in the brain-seeded fibrils of Ghosh et al. are absent from the lipidic fibrils, or because the brain-seeded fibrils were more highly polymorphic?

We have observed that the solid-state NMR spectra of the lipidic fibrils are better resolved than those of the brain-seeded fibrils reported by Ghosh et al. Upon comparing the hCANH spectra of both fibril types, we observed that 9 out of the 21 assigned residues from the brain-seeded fibrils showed a similar chemical shift in the lipidic fibrils, which is V18-F20, E22, V24, I32, G33, G37, and K28'.

The NOE data obtained from the lipidic fibrils reveal acyl-chain cross-contacts at positions K28 to V36, K16 to F19, and D23. In contrast, the brain-seeded fibrils exhibit cross-contacts within the beta-hairpin structure of A β 40, specifically between residues K28 and V40, as corroborated by the cryo-EM density map.

We proposed in the manuscript that these differences in cross-contact between the acyl chain of lipid and fibril contribute to the different chemical shifts in the lipidic fibril.

Furthermore, these differences may contribute to the enhanced spectral resolution observed in the lipidic fibrils. And also, aside from the outer hairpin layer, the paper did not describe a high degree of polymorphism in the brain-seeded fibrils (Ghosh et al.). Consequently, we also suggest that the beta-hairpin layer also could be a potential factor in the reduced spectral resolution observed in brain-seeded fibrils.

In the revised manuscript, we added the following paragraph to the Results and Discussion section on page 4, starting at line 24.

"The high-resolution cryo-EM map allows us to accurately model residues D1 - V40 (Fig. S4a). The L1 fibril is almost identical to previously described A β fibrils derived from the brain tissue of an AD patient by seeded fibril growth³³ (Fig. 2c, d). A comparative analysis of Ca and NH chemical shifts showed that 9 of the 21 residues in

brain-seeded fibrils had congruent chemical shifts in lipidic fibrils (V18-F20, E22, V24, I32, G33, G37, and K28'). The main difference between the fibrils is that in the brain-seeded fibrils, a proteinaceous density is found approximately where lipid abuts the L1 fibril. This proteinaceous density was proposed to be composed of A β 40 beta hairpins based on REDOR data and mass-per-length. Furthermore, we observed additional interactions with the acyl chain (Q15-F19) (Fig 3. b) and the head group (D1-H13) (Fig 2. b) of liposomes in the L1 fibril. This analysis indicates that the variations in chemical shifts arise from brain-seeded fibrils interacting with peptides, whereas in our case, fibrils interact with lipids. Furthermore, in contrast to the structure presented here, the N-terminus of the brain-seeded fibrils was not entirely resolved, suggesting that the lipids may be relevant for partial fibril stabilization."

4. The authors say that the D23-K28 salt bridge "was previously reported...to be populated to around 50%", and they cite the paper by Ghosh et al. I think this is incorrect. The fsREDOR data in the paper by Ghosh et al. seem to indicate nearly full occupancy of the D23-K28 salt bridge.

Thank you for bringing this to our attention. We have reviewed the paper by Ghosh et al. They presented evidence for a D23-K28 salt bridge, as revealed by the cryoEM density map and fsREDOR experiments between the D23 C γ and the K28 N ϵ and molecular models (6W00). This molecular model was calculated from ten independent simulated annealing calculations performed with Xplor-NIH, utilizing the cryoEM density map and NMR data. In that model from the 10 calculations, approximately half of the structure contains the salt bridge and the other half does not.

In the revised manuscript we have edited the following sentence in the Results and Discussion section on page 6, starting at line 4:

"Indeed, according to the molecular model (PDB-ID: 6W00) previously reported, the salt bridge between these two residues appears to be populated at approximately 50%, with the remaining 50% being devoid of this salt bridge.³³"

5. According to Fig. S1, the L1 fibrils represent 63% of the particles. The solid-state NMR spectra indicate a higher level of structural homogeneity. Why? Is this because the relative numbers of cryoEM particles are not an accurate representation of the relative quantities (i.e., masses) of different polymorphs? Or is this because the solid state NMR samples were really more homogeneous than the cryoEM samples? Were the same batches of fibril used for both cryoEM and solid state NMR?

Thank you for your insightful question. We would like to clarify that the samples used for both cryo-EM and solid-state NMR are identical, meaning that the cryo-EM structure was done from the $^{13}\text{C}/^{15}\text{N}$ labeled lipidic A β fibrils.

Regarding the cryo-EM experiments, the determination of the relative population was estimated after 2D classification of ~3 million particle segments, leading to the distribution shown in Fig. S1. 2D classification provides a simple way to sort the extracted particles, and particles sorted into well-resolved classes are then separated from other particles sorted into poor-resolution classes. It also allows for removing particle segments that are contaminated, or that could not be assigned. However, this procedure may struggle to provide a highly accurate representation of the relative distribution of polymorphs, as it depends on the type and amount of particles picked from the micrographs. One may even question whether the fibrils seen on the micrographs indeed represent the distribution of the fibrils in the sample or whether the population has been biased during grid preparation, in particular during blotting and plunge freezing. Nevertheless, in our opinion, this is still the best way of estimating the relative distribution of polymorphs based on cryo-EM data, particularly if one used ~3 million particles. Yet, these caveats indicate that there is no discrepancy between the populations seen in cryo-EM and NMR, especially taking into account the considerations below.

Regarding the character of the NMR spectrum, it represents the homogeneous dominant structure and may not exhibit signals from the non-dominant structure. This is because the spectrum potentially averages out the signals from the smaller population of heterogeneous structures.

In the L1 structural configuration, each particle comprises two molecular entities, thus resulting in a twofold enhancement of the NMR signal intensity. Conversely, the L2 and L3 structures contain only a single molecule per particle. As a result, the cryo-EM data would indicate that the L1 fibrils should account for approximately 78% of the sample, while other polymorphs represent 22% (15% L2+L3, 5% L2/L3, 2% L2/L2+L3/L3). This level of signal (15%, 5%, 2%) would be difficult to detect in the solid-state NMR spectra, which have a signal-to-noise (S/N) level of 21 for L1 in the most sensitive 3D spectrum, the hCANH spectrum. This would leave a S/N level of about 4 for the L2+L3 population, if the signals from L2 and L3 would overlap, which is already difficult to distinguish from noise.

Thus, we conclude that the percentages seen in NMR and cryo-EM are compatible with each other. In the revised manuscript we have extended the corresponding paragraph (on page 5, starting at line 16):

"The following evidence suggests that L1 also dominates the NMR spectra. First, secondary chemical shifts analyzed by Talos N predict the secondary structure of A β 40 fibrils to consist of two β -strands ranging from residues H13 to D23 and I31 to V36 that are connected by a loop region. Second, contact between V24 and G33/L34

observed in DARR spectra is consistent with the inter-strand contacts seen in the cryo-EM structure of L1 fibrils (**Error! Reference source not found.**). Third, contacts are observed from V24-L34, V24-G33, and S26-D23 that are only compatible with the L1 structure. In the case of the L2 and L3 fibril structures, these inter-residue distances exceed 20 Å (V24-G33, L34) and 8 Å (S26-D23). In the L1 fibril, each layer comprises two molecular entities, leading to a twofold increase in solid-state NMR signal intensity. Conversely, L2 and L3 structures consist of only one molecule per layer. As a result, based on the cryo-EM, L1 fibrils can be expected to account for about 78% of the total signal, while other polymorphs contribute the remaining 22% (15% L2+L3, 5% L2/L3, 2% L2/L2+L3/L3).

Based on these signal distribution estimates within the sample and signal-to-noise of about 21 for peaks in the assignment spectra, we conclude that only the L1 polymorph was detectable by NMR. It should be noted that the population estimate from the cryo-EM dataset may not be accurate for several reasons: 1) different polymorphs could react differently to blotting before plunge freezing, 2) automated particle picking might be more effective for the denser L1 fibril, and 3) during image classification only well-defined classes that could be unambiguously identified as one of the polymorphs were selected for further processing."

6. Another possibility is that the L2 and L3 fibrils are less stable thermodynamically than the L1 fibrils, so that the L2 and L3 fibrils could convert slowly to L1 fibrils during or prior to solid state NMR measurements. Did the authors investigate this possibility? Perhaps they should at least mention that the different polymorphs are likely to have different stabilities. I would guess that the the L1 fibrils are more stable than other polymorphs, due to their more extensive intermolecular contacts.

While this is certainly a possibility, we cannot tell from our data if there is a conversion since the NMR measurements take a considerable amount of time. Data collection for the back-bone assignment took approximately 6 weeks. As pointed out in the previous answer, we cannot expect perfect agreement, considering that the cryo-EM reconstructions are made from about 10 percent of the fibrils, and also that detection of minor populations by MAS NMR is challenging. The stability/evolution of fibril structure would indeed be an interesting topic to explore in future work. Without data regarding fibril evolution, we would prefer not to comment on it here.

Reviewer #2 (Remarks to the Author):

In this manuscript, the authors report the polymorphic structures of A β 40 fibrils grown with lipid membranes by using cryo-EM. They provide detailed structural information about the fibrils and compare them with previously reported A β 40 fibrils conformations. As lipid molecules are considered critical factors in modulating protein aggregation, this study will provide insights into protein-lipid interactions during amyloid fibril formation, but some issues listed below should be addressed before publication.

1. Cellular membranes are formed by various lipids, but in this study, the authors prepared the vesicles using a saturated and negatively charged DMPG lipid. Why did the authors choose DMPG as a model system to study the fibril-lipid interactions? Are there any specific reasons for this, such as the important physiological roles of DMPG in Alzheimer's disease and Ab aggregation?

Thank you for your query regarding our choice of DMPG as a model lipid system for studying fibril-lipid interactions. There is no specific evidence indicating a direct physiological role of DMPG in Alzheimer's disease. However, many previous studies reported that negatively charged liposomes induced A β aggregation, especially A β 40 aggregation (for example, reviewed in Rangachari et al., *Biochim. Biophys. Acta Biomembr.*, 2018.)

The negatively charged liposomes provide a simplified model system for the aggregation of A β 40, but it does not fully replicate the complexity and diversity of lipids in neuronal cell membranes. Our choice of DMPG was its transition temperature (T_m) of 24°, which allows us to use approximate physiological conditions of neuronal cell membranes, which are also fluid in the A β 40 aggregation temperatures.

In the present study, we employed a low pH of 6.5 to induce protonation of the three histidine residues in the monomeric, non-fibrillar A β 40 peptide. These protonated histidine residues interacted with the negatively charged head groups of DMPG. Such interactions foster the lipid-mediated aggregation of A β 40.

In the revised manuscript we have included the following paragraph on page 7, line 21:

"In the present study, we utilized a mildly acidic pH of 6.5 to facilitate partial protonation of the three histidine residues in the monomeric, non-fibrillar A β 40 peptide. This facilitates enhanced electrostatic interactions between the A β 40 monomer and negatively charged head groups of liposomes and promotes A β 40 aggregation. For the study, DMPG was selected as the lipid due to its transition

temperature (T_m) of 24°C, approximating the physiological conditions experienced by neuronal cell membranes at our designated incubation temperature of 37°C."

2. The authors mentioned that "Initial characterization by negative stain EM revealed fibrils longer than 1 μm , most of them being in contact with spherical or incomplete fibril-attached features (Fig. 1a, b)". However, it is hard to find what are the incomplete fibril-attached features in the figure, so it would be good to add arrows to indicate these features. And also, please add descriptions about yellow arrows in Fig 1b.

We apologize for any confusion that may have arisen from our mistake. After re-evaluating the text, we found an error during the editing process. As per your suggestion, we have now amended the sentence in question to read:

"Initial characterization by negative stain EM revealed fibrils longer than 1 μm , most of them being in contact with spherical or incomplete liposomes attached to the fibril features (Fig. 1a, b)."

Upon thorough examination of the negative-stain EM images, we were unable to observe any incomplete liposomes associated with the fibrils potentially due to the preparation conditions (drying and staining). However, in the Cryo-EM images of the same sample used for negative stain EM and solid-state NMR, we were able to observe incomplete liposomes associated with the fibrils, which are indicated by the yellow arrows in Fig 1b. We apologize for the confusion and added a description to the figure caption (see page 22), which now reads:

"Fig. 1 | A β 40 fibrils in the presence of phospholipids.

a, b, A negative stain transmission electron microscopy (TEM) micrograph (**a**) and a 20 Å low-pass filtered cryo-electron microscopy (cryo-EM) micrograph (**b**) of lipidic A β 40 fibrils. The yellow arrows indicate the fibril-bound incomplete liposomes. **c**, The sequence of A β 40. Residues assigned by NMR are colored blue. **d**, Examples of 2D class averages for L1. The yellow arrows indicate the fibril-bound layers of lipids (incomplete liposomes), lacking the characteristic amyloid cross- β pattern, while the cross- β pattern is evident for the A β 40 fibrils. **e**, 3D(H)CANH spectrum of lipidic A β 40 fibrils. Red arrows depict the two populations of K28 and D23."

3. The samples are prepared by mixing Ab proteins with lipid vesicles, but negative stained EM and raw micrograph images only showed the fibril without lipid vesicles. Did authors observe vesicle-associated fibrils or broken vesicles by fibrils from negative stained EM or cryo-EM images? Since the reconstructed 3D density maps show lipid features, there are probably cryoEM images that show membrane-associated fibrils. If yes, please show the images. I think this will help readers to understand how lipid vesicles interact with fibrils.

We inspected all micrographs one more time but did not observe an intact or at least partially intact liposome. Instead, the micrographs show the fibrils, which are decorated with lipids (yellow arrows in Fig. 1b), but also contrast-rich, bilayer-similar objects, which we assumed to be parts of the initial liposome that reshape during lipid extraction from the liposomes during fibril growth. A possible explanation is that fibril growth is associated with lipid vesicle disruption and remodeling.

We have added a discussion of such a potential mechanism at the end of our manuscript. See also reply to question 8.

4. In this study, cryo-EM report three different folds, and the secondary structure of L1 significantly differs from L2 or L3. For example, the C-terminal region of L1 is different from the same region of L2 and L3 polymorphs. Therefore, readers may expect the heterogeneous ssNMR spectra (multiple peaks) because chemical shifts of NMR spectra are known to be very sensitive to the secondary structure of the protein. However, the peaks in the ssNMR spectra are very homogeneous except for residues D23 and K28. Can the author provide more explanation about this, why signals from L2 and L3 polymorphs are invisible in ssNMR spectra except D23 and K28?

This question is similar to question 5 and 6 by reviewer #1, please find the answer there.

5. L2 fold has a fibril core from residue 5 to residue 40, which means there is a somewhat mobile N-terminal region (residue 1-4). Have the authors tried to measure the dynamic signals by NMR?

Investigating the dynamics of the protein could indeed be an interesting topic. We chose not to investigate dynamics here, including searching for dynamic signals using j-coupling-based pulse sequences. Instead, our focus was on integrating cryo-EM data and NMR data for the structured regions of the fibril. In our experience and considering the correlation length of IDPs should be long enough, the 4 residues are unlikely to be sufficient to observe intense signals in j-coupling-based spectra. Additionally, it is important to note that in all of the structures discussed in this report, the N-terminal 4/3/2/1 residues are in close proximity to the head group of liposomes. These interactions may further slow dynamics.

6. The figure number should be corrected. There are two Fig. 1 on pages 18 and 19, so all figure numbers are not matched with the main text.

Thank you for bringing this to our attention. We regret the oversight and we have corrected the figure numbers.

7. Reference #36 page 4 seems wrong. I think the author wanted to refer to #34, not #36. However, the authors are correct and intend to refer to #36, disregard my comment.

Thank you for bringing this to our attention. This has been corrected in the revised manuscript.

8. In conclusion, the authors claim that “The complex structures provide a molecular-level structural understanding of how A β aggregation can lead to lipid extraction from vesicle”. How do authors conclude that lipids associated with fibrils in density maps are extracted from vesicles? I think sonicated vesicles may be simply attached to the fibrils. Are there any further data to support lipid extraction from vesicles?

Thank you for your inquiry concerning liposome behavior during A β 40 aggregation.

First, let us clarify the methodology used for A β 40 fibril formation. In preparing the liposomes through sonication, we ascertained that the average liposome diameter exceeded 80 nm, as verified by dynamic light scattering (DLS) measurements.

In Figure 1a, we observed a spherical-shaped liposome on the fibril, exhibiting a diameter of around 100 nm, as you mentioned. During aggregation, no sonication was applied, such that the less than 80nm lipid assemblies can be attributed to their interaction with aggregating A β 40. The micelle-shaped lipid assemblies in the density map, which have a diameter of around 10 nm, are smaller or similar to that of a single A β 40 fibril molecule (length of L1: 13.4 nm, L2: 5.1 nm, L3: 6.6 nm). The density maps are presented in Figures 2a, 4a, 5a, b, and c. In the case of L1, an extra density is located between Lys28 and Val40 and the distance between these residues was measured at 3.3 nm, corresponding to the combined length of two acyl chains in a DMPG lipid molecule. The two acyl chain extra densities are presented in Figure 2a. The NOE data presented in Figure 3b corroborates this observation.

Since lipids started in liposomes, but some of these were eventually found in micellar shape lipid assemblies associated with fibrils, we concluded: "The complex structures provide a molecular-level structural understanding of how A β aggregation can lead to lipid extraction from the vesicle."

In the revised manuscript we have included the following paragraph (on page 7, starting at line 21):

“In the present study, we utilized a mildly acidic pH of 6.5 to facilitate partial protonation of the three histidine residues in the monomeric, non-fibrillar A β 40 peptide. This facilitates enhanced electrostatic interactions between the A β 40 monomer and negatively charged head groups of liposomes and promotes A β 40

aggregation. For the study, DMPG was selected as the lipid due to its transition temperature (T_m) of 24°C, approximating the physiological conditions experienced by neuronal cell membranes at our designated incubation temperature of 37°C.

For the aggregation studies, liposomes were prepared using a sonication method. Dynamic light scattering (DLS) measurements confirmed that the average diameter of these liposomes exceeded 80 nm. After aggregation, we observed the presence of spherical-shaped liposomes on the fibrils, each with an approximate diameter of 100 nm (Fig. 1a), as evidenced by negative-staining electron microscopy (EM) data. It is noteworthy that during the aggregation process, no sonication was applied; thus, any observed lipid assembly with a diameter less than 80 nm can be attributed to their interaction with aggregating A β 40.

In our density map analysis, we observed lipid micelles on the A β 40 fibril molecule with a diameter of approximately 10 nm, smaller or similar to an individual A β 40 fibril molecule (dimensions L1: 13.4 nm, L2: 5.1 nm, L3: 6.6 nm). In the case of L1, extra density is located between Lys28 and Val40, extending over 3.3 nm, corresponding to the combined length of two acyl chains in a DMPG lipid molecule (Fig2. a, b). This observation is corroborated by the NOE data (Fig. 3b). Altogether, we propose that during A β 40 aggregation, lipids interact with A β 40 and form such lipid micelles on the surface of the fibrils.”

Reviewer #3 (Remarks to the Author):

The authors have reported cryo-EM structures of amyloid-beta(1-40) fibrils in lipid vesicles, which might be useful in investigating fibril-lipid interactions present in Alzheimer's patients. The authors have also used MD simulations to investigate hydrogen bond interactions between D23 and K28. I only have the following questions regarding the MD simulations for the authors:

1. Please state the pH (this is usually set with PROPKA or H++ programs) and the ionic concentration that were used for the MD simulations. I'm a bit concerned that the right pH and ionic concentrations were not set since they were not stated in the methods. Please also state whether a cubic or octahedral box was used for the solvent.

We apologize for the confusion. For MD simulations, we considered pH 7. We only added 72 Na⁺ sodium ions to neutralize the entire system and did not add additional ions.

Please note that during model building, we use *PROPKA-3* to predict the pKa and protonation states of titratable side-chains to evaluate whether side-chains are forming reasonable interactions or whether further structural refinement of the side-chain rotamers is desirable. However, as the resolution of the cryo-EM maps in this study does not allow for the modeling of the positions of hydrogen atoms (Yip, *et al.*, Nature, 2020, Nakane, *et al.*, 2020), the final models do not contain the predicted protonation states and *PROPKA-3* is considered one of many intermediate steps towards the final model.

For the L1 fibril, the chain-wise (A-J) *PROPKA-3*-predicted pKa values read:

Res. ID	A	B	C	D	E	F	G	H	I	J	AVG
E 3	4.71	4.71	4.99	4.99	5.31	5.31	5.01	5.01	4.70	4.70	4.94
R 5	14.0	14.0	13.7	13.7	13.3	13.3	14.0	14.0	12.4	12.4	13.5
	6	6	4	4	8	8	3	3	0	0	2
H 6	6.64	6.64	5.95	5.96	8.45	8.46	5.32	5.32	7.03	7.02	6.68
D 7	3.44	3.44	2.14	2.14	3.14	3.14	2.61	2.61	1.93	1.92	2.65
Y 10	10.2	10.2	10.6	10.6	11.0	11.0	10.6	10.6	10.2	10.1	10.5
	1	5	1	9	8	9	5	1	0	7	6
E 11	4.52	4.57	2.63	2.64	7.49	7.48	4.19	4.15	4.54	4.54	4.68
H 13	6.36	6.37	5.48	5.45	4.04	4.04	4.84	4.85	6.41	6.44	5.43
H 14	6.11	6.11	5.74	5.73	5.31	5.31	5.70	5.70	6.04	6.04	5.78
K 16	10.3	10.3	10.0	9.98	9.67	9.67	9.98	10.0	10.3	10.3	10.0
	6	6	0					0	6	6	7
E 22	3.74	3.75	4.39	4.47	5.27	5.28	4.66	4.63	3.97	3.96	4.41
D 23	1.32	1.51	2.62	2.75	3.68	3.68	2.28	2.17	2.80	2.60	2.54
K 28	12.3	12.3	13.8	13.8	12.9	13.0	13.4	13.4	13.8	13.7	13.3
	9	9	6	5	9	0	8	6	2	9	0

For the MD simulations, the protonation states were then assigned based on the average (AVG) pKa values, such that at pH 7, all glutamate and aspartate residues are negatively charged, lysine and arginine positively charged, and tyrosine and histidine neutral. Of particular interest to us are the protonation states of D23 and K28, which are both charged at pH 7. One might consider H6 partially protonated and positively charged at pH 7, but considering its separation from D23 and K28, it will not affect this interaction, in particular, as we restrained the backbone, which makes any relocation of protein domains impossible.

Also, considering the reviewer's concerns about the shape of the simulation box, we rephrased the respective Method section to be more precise. The new section "*Molecular dynamics simulations of the L1 A β 40 fibril.*" (page 12, line 8) now reads:

"Residue protonation states were assigned for a pH of 7 and hydrogen atoms were added to the cryo-EM structure according to the Amber ff19SB force field library⁵⁴ by using LEaP⁵⁵, which is distributed with the Amber 22 suite of programs (comprised of AmberTools22 and Amber22)⁵⁶, such that all glutamate and aspartate residues are negatively charged, lysine and arginine positively charged, and tyrosine and histidine neutral. After adding 72 sodium ions to neutralize the system, we placed the fibril-ion complex into a truncated-octahedron solvent box leaving at least 15 Å between any solute atom and the edge of the simulation box. The Amber ff19SB force field⁵⁴ was applied to describe the A β 40 fibril. Ion parameters for sodium ions were taken from ref. ⁵⁷ and used in with the OPC water model⁵⁸."

2. Ten simulations with 1 microsecond long each seem to be long enough but it would be better to justify the simulation length as done in the SI of Robustelli et al. JACS 2022 (i.e., find the autocorrelation times).

Following the reviewer's suggestion, we calculated the autocorrelation functions of the hydrogen bond interactions between D23 and K28, for each replica simulation individually (**Fig. S13a**) and for the aggregated trajectory (10 x 1 μ s) (**Fig. S13b**). The data is shown in the new Fig. S13 (page 42). The autocorrelation analysis revealed major bumps within the first 400 ns and smaller bumps in during the 400 – 800 ns interval. After 800 ns, only a few individual interactions show minor fluctuations. Considering the aggregated trajectory, the bumps are smaller throughout the full simulation time, with only minor variations after 200 ns. We inserted the following paragraph in the methods section:

"We also calculated the autocorrelation function of the hydrogen bond interactions between D23 and K28, for each replica simulation individually (**Error! Reference source not found.a**) but also for the cumulative trajectory (10 x 1 μ s) (**Error! Reference source not found.b**)."

3. It's unclear how the conformations were chosen and extracted for Fig S11. Was a clustering method used or were these final snapshots at 1 microsecond for each of the ten simulations? It would be more informative if a clustering analysis was done to show that these certain highly mobile or immobile conformations are the dominant conformations depending on the restraints.

In the original version of the manuscript, Fig. S11 showed the final conformation after 1 μ s for each of the ten simulations. Now, following the reviewer's suggestion, we clustered all conformations by the $C\alpha$ -RMSD. In Fig. S11b we show the representative structures of the top five clusters. In line with the results from Fig. S11, pronounced structural changes are only observed in the absence of positional restraints.

In addition, we now also show the $C\alpha$ -RMSD relative to the cryo-EM structure in the new Fig. S12 for simulations with and without additional weak restraints. In the absence of the restraints, the RMSD increases to ~ 10 Å, while in the presence of restraints the RMSD remains at ~ 1.2 Å. Together with the structures shown in Fig. S11, this data supports the need for weak positional restraints to stabilize the fibrillar model.

In the revised manuscript, we also added the clustering to the Methods section on page 13, starting at line 12.

"For the hierarchical agglomerative clustering of the non-restrained trajectories, the distance between the coordinate frames was calculated via a best-fit coordinate RMSD considering all $C\alpha$ atoms. The clustering was finished when the minimum distance ϵ between clusters was greater than 4 Å."

REVIEWERS' COMMENTS

Reviewer #1 (Remarks to the Author):

In this revised manuscript, the authors have addressed my original comments adequately. It seems to me that they have also addressed the comments of other reviewers adequately. Therefore, I recommend in favor of publication in Nature Communications without further review. This is an excellent paper.

Reviewer #2 (Remarks to the Author):

The authors have responded to all my questions and modified the manuscript sufficiently. I support publication in this current version.

Reviewer #3 (Remarks to the Author):

Thank you for addressing my concerns. No further changes are required regarding the MD simulation sections.